# Attitudes of blood and plasma donors towards biobanking for longitudinal health research

**Karlijn F. Joosten**[1]*, **Lotte van Dammen**[1], **Eva-Maria Merz**[1,2], **Věra M. J. Novotný**[3], **Daphne C. Thijssen-Timmer**[4], **Katja van den Hurk**[1,5]

**1** Department of Research, Sanquin Blood Supply Foundation, Amsterdam, The Netherlands, **2** Department of Sociology, Vrije Universiteit Amsterdam, Amsterdam, The Netherlands, **3** Department of Medical Affairs, Sanquin Blood Supply Foundation, Amsterdam, The Netherlands, **4** Executive Committee, Blood Bank, Sanquin Blood Supply Foundation, Amsterdam, The Netherlands, **5** Department of Public and Occupational Health, Amsterdam Public Health (APH) Research Institute, Amsterdam UMC, Amsterdam, The Netherlands

* kfjoosten@hotmail.com

## Abstract

### Background

Blood and plasma donations provide a unique opportunity for setting up a biobank. Remainders of donations can be stored together with questionnaire data to allow for longitudinal health research. Insight in the attitude and understanding of potential concerns of donors towards biobanking allows incorporating their perspective in the set-up of biobanks as such.

### Methods

In an online survey in July/August 2021 among 409 Dutch donors, we asked about their attitudes towards the collection and storage of remainders of donations, questionnaires, and DNA, and towards the storage period, data sharing and linkage, and feeding back results and outcomes.

### Results

The overall attitude of participating donors towards biobanking remainders of donations is very positive with 99% indicating willingness to provide their consent and 94% to fill out questionnaires. Most respondents (74%, n = 292/395) would agree to a long-term storage of 30 years. When asked about potential concerns, respondents mostly mentioned personal data security (30%), commercial use of data (11%) and misuse of data (5%). Also, 40% (n = 155/390) showed some hesitance towards the storage of DNA, including 28% (n = 111/390) who would give conditional consent if being informed on for instance DNA utilisation. Finally, the majority would like to receive feedback of research results, and indicated this to contribute to a greater likelihood of biobank participation.

**Data availability statement:** All relevant data are within the paper and its Supporting Information files.

**Funding:** This research project was funded by a Sanquin Product and Process Development – Cellular Products Grant, project id: PPOC20-09/ L2494, received by K. van den Hurk. The funders had no role in study design, data collection and analysis, decision to publish, or preparation of the manuscript.

**Competing interests:** The authors have declared that no competing interests exist.

## Conclusion

Our findings show high support among surveyed donors for the storage of donation remainders in a biobank, provided high data security standards and clear communication about data protection measures and usage of biobank data and materials.

## Introduction

Blood and plasma donation is of inestimable value in health care, as transfusions are required as supportive care in several diseases alongside their use in case of severe blood loss. Besides that, there are increasing demands for plasma-derived medicinal products [1]. With the repetitive nature of voluntary blood and plasma donation comes a unique opportunity to use (remainders of) donated blood and plasma to set up a biobank with healthy participants, as opposed to existing patient biobanks. Compared to patient biobanks, healthy participant biobanks represent a less selective group of people and can provide more information about population health. Furthermore, healthy participant data in disease-genetics studies prevent the confounding of treatments or disease progression on the genetic variants of interest. Samples from multiple donations per individual donor can be stored for many years for scientific research purposes. Research on these samples, supplemented with questionnaire and linked registry data, may help to improve our understanding of transfusion medicine, as well as public health and disease development. The large-scale biobank study FinnGen [2] has for example shown the possibility to discover genetic variants to be related to certain diseases by combining genetic data with data from health registries. Multiple studies and existing biobanks have used this principle of collecting blood and data alongside standard blood donations for research purposes [3–6].

Before setting up a donor biobank, it is important to assess the attitudes of potential participants towards the collection, storage and usage of questionnaire data and blood samples [7,8]. Attitudes include thoughts, feelings, and concerns towards in this case a biobank. Opinions and concerns that may deter from participation can then be taken into account in setting up the biobank and communication towards participants. Blood banks in other countries have assessed the attitudes of their donors to biobanking and their willingness to provide blood samples for biobank storage. The most comprehensive study in this field was carried out in Australia [9]. More than 95% of the survey respondents (n = 827) indicated their willingness to provide blood samples for research, and over 90% of this group would give consent for health research involving genetic testing and for the linking of data with external database information [9]. Likewise, a study among 616 donors from the Gaza Strip showed a positive attitude towards the use of donated blood for medical research and the linking of health information to blood samples among respectively 86% and 64.5% of the donors [10]. In a Finnish qualitative study (n = 61) regarding willingness to donate samples for biobanking purposes, results were mostly positive, with some concerns regarding protection of personal data [7].

In the Netherlands, the Netherlands Patients Federation performed a general study on Dutch patients' attitudes towards the use of leftover biospecimen in hospitals for research purposes. They found a positive attitude among 97% of 12 300 questionnaire participants [8]. Contrary to patients, donors are usually healthy, which may limit their perceived personal benefit of the use of blood materials for research purposes. Therefore, donors' attitudes towards biobanking may differ from those of patients.

Within the Netherlands no specific studies on attitudes towards biobanking have been carried out among donors. Attitudes towards donor biobanks in other countries [7,9,10] may differ due to cultural, societal and institutional differences, and differences in levels of trust towards the health care system and blood collection agencies. Furthermore, previous studies have not assessed whether donors, in contrast to patients, wish to be informed about results and research outcomes. Therefore, the current study aims to investigate the attitude of Dutch blood and plasma donors (from now on referred to as blood donors) towards the storage of samples, data, and DNA, storage periods, and data sharing and linking. In addition, this study aims to provide insight in if, and in what form, donors wish to be informed about results and research outcomes when participating in a biobank. A survey among Dutch blood donors provided insight into all of the above.

## Materials and methods

### Overall approach

To assess the attitude of donors to storing leftover blood materials and data in a biobank, data sharing, DNA storage, storage period and their opinions about potential feedback on research results, a cross-sectional online survey was performed among blood donors. The survey was active for five weeks from 29 July to 2 September 2021.

### Participants

The donor population in the Netherlands, donating at Sanquin Blood Bank, comprised 391 288 blood donors in 2021 [11]. Using the sample size (n) calculation of $n = \frac{N \cdot z^2 \cdot p(1-p)}{e^2(N-1) + z^2 \cdot p(1-p)}$, with population size (N) = 391 288 blood donors, Z-score (z) = 1.96 for a confidence level of 95%, the population proportion (p) = 0.5 for maximum variability, and the margin of error (e) = 0.05, we aimed to include at least 384 donors in this survey: $n = \frac{391288 \cdot 1.96^2 \cdot 0.5(1-0.5)}{0.05^2(391288-1) + 1.96^2 \cdot 0.5(1-0.5)} \approx 384$. To collect responses to the survey, an invitation to the open survey was embedded in the Sanquin email newsletter, sent monthly to 170 000 subscribers. The survey invitation was sent to a randomly selected group of 10 000 subscribers, using the marketing automation platform Copernica (Copernica, Amsterdam, the Netherlands), on 29 July 2021. A second invitation was sent with the Sanquin newsletter on 26 August 2021, addressing specifically another 1000 donors who were not previously approached for the survey. The only exclusion criterion, as assessed in the online informed consent, was age below 18 years, in accordance with the donation criteria in the Netherlands.

### Data collection

An online survey was chosen to allow for easy distribution among a large group of donors. The survey was programmed using Qualtrics software (Qualtrics, Provo, UT) with free navigation, and tested on usability and functionality. Before completing the survey, participants filled out a compulsory, digitally written, informed consent form to obtain permission to use their anonymous information.

The survey included seven questions covering demographic variables. This way, the participants could be categorised by gender, province of residence, donation history, and predefined groups for age and educational level (S1 File). All survey questions about biobanking were based on previous biobanking studies [7,8]. Our survey included 16 multiple choice questions regarding biobanking. The first five questions covered the participants' attitudes and ideas towards storing leftover blood materials and data in a biobank, followed by questions regarding data sharing and linking, DNA

storage, storage period, and opinions about potential feedback of research results. The responses yielded a quantitative assessment of the attitudes and concerns of donors. Furthermore, open answer options were programmed in 12 of the multiple-choice questions, and two open-ended questions were included at the end for participants to write down their biggest concerns and potential questions or remarks, to complement quantitative data insights.

## Statistical analyses

Descriptive statistics (counts, percentages, etc.) were used to describe results of the survey. The percentages were rounded to the nearest whole number, whereas numbers between zero and one were rounded to one decimal. The free text answers in the 'other' fields and answers provided for the last two open questions were coded, using in vivo coding and pattern coding [12], and described.

The 426 registered survey responses, distinguished by unique codes per IP-address, included 43 unfinished responses, of which 11 were removed that were not progressed up to and including question five of the biobanking questions. Furthermore, survey responses from non-donors (n = 6) were removed, resulting in a total of 409 responses left for analysing. Further adjustments to the data comprise recoding open answer options corresponding to one of the multiple-choice options, recoding multiple checked multiple-choice options to one overarching multiple-choice option in case of redundancy, and omitting any arguments in the "Other (please specify)" option open field that related to a different question.

## Ethics statement

This study was conducted according to the principles of the Declaration of Helsinki [13]. Participants were not exposed to any risks or harm in this study. The Executive Committee of Sanquin has provided permission to carry out the study, thereby agreeing that the study concerns a short, anonymous questionnaire that does not require a *Medical Ethical Committee* assessment because it does not fall under the scope of the Medical Research Involving Human Subjects Act (WMO) [14].

The survey was fully anonymous as no identifying information was collected. Participants were required to provide their informed consent prior to completing the survey. Participation in the study was on a voluntary basis, without incentives offered, and without any further consequences.

## Results

### Overview of study sample: Demographics

The responses of 409 participants were included in the analyses. The demographic characteristics reflect donors from all predefined age groups, with an equal male/female ratio (Table 1). The respondents have varied educational backgrounds and represent all the different provinces of the Netherlands, with an overrepresentation of donors from more urban provinces.

### Biobanking and questionnaires

We asked the participants about their knowledge of the term biobank; 73% had never heard of a biobank or was not sure, while 27% was familiar with the term or had heard of it (Table 2). Upon providing the biobank definition, the vast majority (99%) of participants indicated that they would consent to collect and use remainders of their donations in a biobank. Furthermore, 94% of the participants would be willing to fill in questionnaires concerning their health and lifestyle when participating in a biobank. No willingness to provide consent for collecting donation remainders or filling in questionnaires was reported by respectively 1% and 1% of the participants.

Among the multiple positive considerations for respectively biobank participation and filling in questionnaires, the most frequently mentioned considerations were 'importance to contribute to science' (respectively 91% and 82%), 'to

**Table 1. Study sample demographic characteristics.**

| Characteristics | Participants (n = 409) |
|---|---|
| **Age**, *n (%)* | |
| Younger than 25 years old | 1 (0.2%) |
| 25–35 years old | 9 (2%) |
| 36–45 years old | 26 (6%) |
| 46–55 years old | 74 (18%) |
| 56–65 years old | 152 (37%) |
| Older than 65 years old | 147 (36%) |
| **Gender**, *n (%)* | |
| Male | 207 (51%) |
| Female | 201 (49%) |
| Other | 1 (0.2%) |
| **Educational level**, *n (%)* | |
| None | 0 (0.0%) |
| Elementary school | 2 (0.5%) |
| Lower vocational education | 32 (8%) |
| Secondary education | 42 (10%) |
| Secondary vocational education | 100 (25%) |
| Higher secondary education | 45 (11%) |
| Higher vocational education | 124 (30%) |
| University | 63 (15%) |
| *Missing* | 1 (0.3%) |
| **Provinces**, *n (%)* | |
| Drenthe | 13 (3%) |
| Flevoland | 8 (2%) |
| Friesland | 12 (3%) |
| Gelderland | 23 (6%) |
| Groningen | 8 (2%) |
| Limburg | 24 (6%) |
| North Brabant | 29 (7%) |
| North Holland | 191 (47%) |
| Overijssel | 16 (4%) |
| South Holland | 44 (11%) |
| Utrecht | 36 (9%) |
| Zeeland | 4 (1%) |
| Other: Belgium | 1 (0.2%) |

help others' (respectively 80% and 77%), 'waste to throw away residual blood' (56%), and 'small effort' (respectively 52% and 55%). Negative considerations for filling in questionnaires included 'unsure what the questionnaires entail' (10%), and for biobank participation 'doubts about secure treatment of personal data' (6%), and 'more information is needed about what happens with the material and data' (5%). Likewise, on the open-ended question concerning their biggest objections and concerns regarding the biobank, the topics of 'security and anonymity, privacy, and careless use of the data' (30%), 'commercial use/interests' (11%), and '(future) misuse of data by, for example, insurance companies' (5%) were frequently mentioned. On the other hand, 19% of the participants indicated to have no objections or concerns in this open question.

**Table 2. Willingness to participate in biobanking and questionnaires.**

| Willingness to participate | Participants (n = 409) |
|---|---|
| **Prior knowledge of the term biobank**, *n (%)* | |
| 'I am familiar with it' or 'I have heard of it' | 112 (27%) |
| 'I am not sure' or 'I have never heard of it' | 297 (73%) |
| **Consent to storage and use of residual blood in a biobank**, *n (%)* | |
| Definitely | 356 (87%[a]) |
| Probably | 47 (11%[a]) |
| I do not know | 2 (0.5%) |
| Probably not | 1 (0.2%) |
| Definitely not | 3 (0.7%) |
| **Consent to fill in questionnaires during biobank participation**, *n (%)* | |
| Definitely | 258 (63%) |
| Probably | 126 (31%) |
| I do not know | 21 (5%) |
| Probably not | 4 (1%) |
| Definitely not | 0 (0%) |

[a] Percentages may not add to 99% due to rounding.

## Storage period, sharing, and linking of bodily material and data

Storing the collected material and data for 30 years would be acceptable for 72% of participants, while 22% answered to provide consent for 15 years, and only 1% would not give consent to store material and data at all (Table 3). Among the 20 responses (5%) with 'Other', the answers included consent for eternity or as long as necessary (2%), and consent for 10 years after which consent should be re-asked (1%).

Furthermore, 62% would provide consent to share biospecimen and data with others both inside and outside of Europe, 28% only within the Netherlands, and 7% only within Sanquin. Among the 'Other' responses (4%) was 'only within the EU' (2%) a recurring reaction.

Respondents were divided about to whom biobank material and questionnaire data should be shared with. Where 95% would give consent to sharing material and data with researchers of organisations active in medical scientific research and 49% to sharing with non-commercial businesses, only 7% indicated to accept sharing with all businesses including commercial enterprises. Among the 2% that indicated 'Other', it was mentioned that sharing with commercial businesses would only be consented under certain conditions, such as open science and profit sharing (1%). Finally, 2% would give no consent at all to share the data.

Linking of biobank data to their medical records would be consented by 60% of the participants. Linkage with other registries, such as Statistics Netherlands and data on environmental exposures, would be accepted by respectively 46% and 41% of participants. Only 10% would allow linkage to information from apps or other online records. Twenty-five % of the participants would not allow any linkage of biobank data.

## Storage of DNA

Eighty-seven % of participants would give consent to store their DNA, including 28% who would only give consent under certain conditions (Fig 1). Eleven % would not give consent. The main conditions under which storing of DNA would be acceptable were 'being kept informed of what my DNA is used for' (66%) and 'if results, like those related to serious, hereditary disease, are communicated to me' (69%).

**Table 3. Willingness to provide consent.**

| Willingness to provide consent<br>*For the two last questions it was allowed to provide multiple answers.* | Participants (n = 395)<br>*Missing n = 14* |
|---|---|
| **Consent on storage period**, *n (%)* | |
| Bodily material and data may be stored for 30 years | 284 (72%) |
| Bodily material and data may be stored for 15 years | 86 (22%) |
| I would not consent at all | 5 (1%) |
| Other | 20 (5%) |
| **Consent to sharing of data with researchers in other countries**, *n (%)* | |
| My bodily material and data may be shared with others (both in and outside Europe) as long as my privacy is protected | 243 (62%) |
| My bodily material and data may be shared with others (in the Netherlands) as long as my privacy is protected | 111 (28%) |
| My bodily material and data may not be shared with others (outside of Sanquin) | 27 (7%) |
| Other | 14 (4%) |
| **Consent to sharing material and data among researchers in other companies and organisations**, *n (%)* | |
| Researchers that work at the blood bank, hospitals, universities, or other organisations that are active in the area of medical scientific research | 374 (95%) |
| Non-commercial businesses | 192 (49%) |
| Businesses, even if they are commercial enterprises | 27 (7%) |
| I would not give my consent at all | 7 (2%) |
| Other | 9 (2%) |
| **Consent to linking of data to other records**, *n (%)* | |
| Biobank data may be linked to information from medical records | 236 (60%) |
| Biobank data may be linked to information from registries (like Statistics Netherlands) | 181 (46%) |
| Biobank data may be linked to information from other sources (like information on my residential area, related to stores, nature, air pollution, etc.) | 161 (41%) |
| Biobank data may be linked to information from apps or other online registries (like activity trackers, supermarket purchases, etc.) | 39 (10%) |
| Biobank data may not be linked to other sources of information | 100 (25%) |
| Other | 10 (3%) |

## Feedback of research results

Considering feedback of research results performed with biospecimen and data from donors, 64% of participants indicated that they would like to receive a yearly newsletter with general research results, and 74% would like to receive an individual report with results pertaining to their health (Fig 2). Only 5% would not like to receive any information. The most prevalent response among 'Other' (4%), was the preference to receive an individual report only when it is medically necessary or relevant (2%). Of those participants who would like to receive an individual report, 39% indicated to be more likely to participate in biobank research by the option to receive an individual report.

## Discussion

This study improves our understanding of Dutch blood donors' attitudes towards storing of their samples and data in a biobank for health research. Overall, donors' attitudes towards participation in such a biobank is very positive, with 99% of participating donors indicating to be definitely/probably willing to provide their consent and 94% also willing to complete questionnaires. Moreover, the majority of donors agrees with storing materials and data for 15 or 30 years as intended for

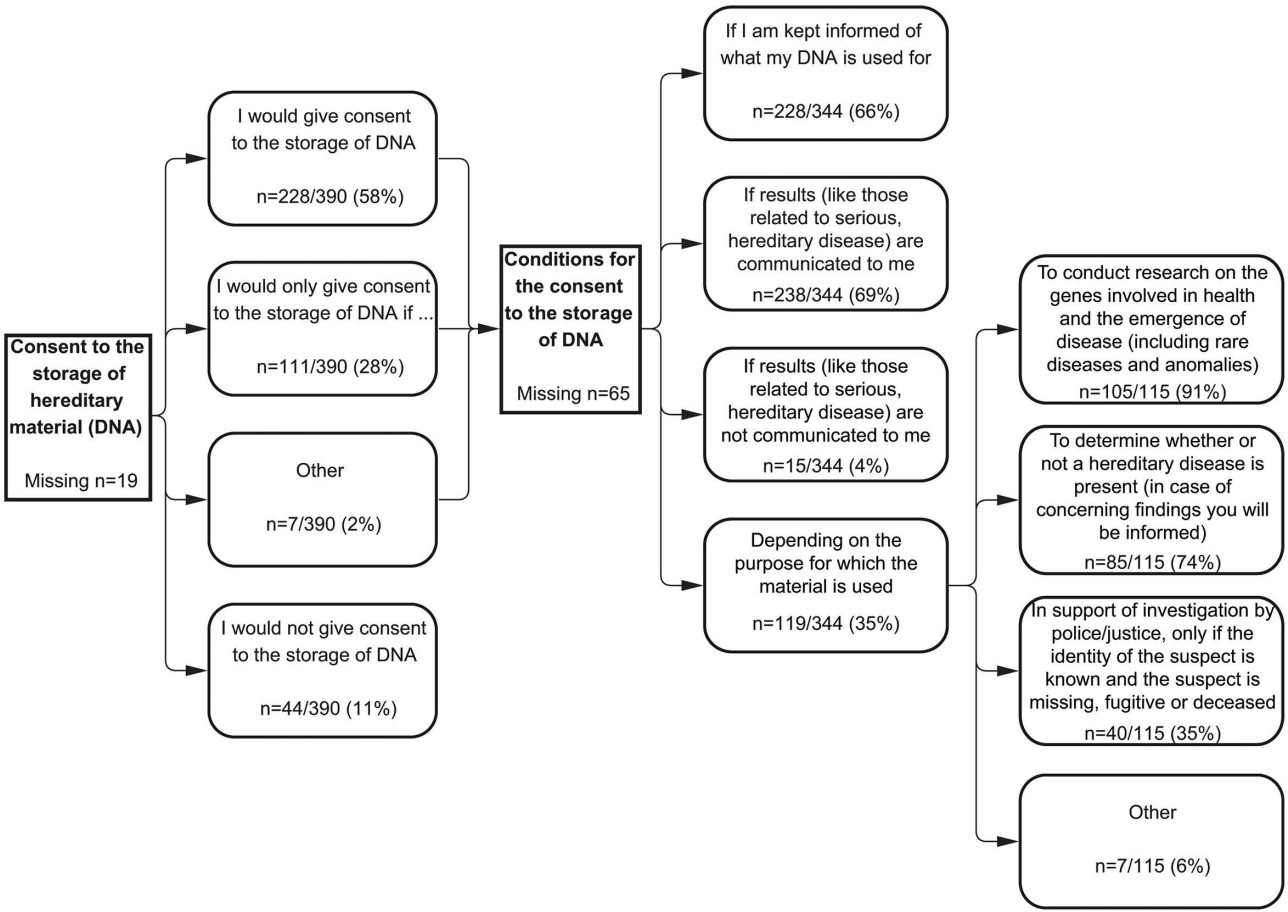

**Fig 1. The consent of participants to store hereditary material (DNA), and the conditions under which participants would provide consent.**
Presented in a flowchart to indicate which responses led to which follow-up questions. Multiple answer options were possible for the two questions regarding the conditions under which consent would be provided.

any biobank. Main concerns as mentioned upon request were the secure treatment of personal data, potential commercial use of data and misuse of data by for example insurance companies.

The high willingness (≥94%) of participants in this study to the use of samples and data for biobanking corresponds with results of previous studies [7–10]. Also, the concerns regarding misuse of data and potential commercial profit match the results of previous work by the Finnish Red Cross Blood Service [7]. Surprisingly, around 73% of respondents was initially not familiar with the term biobank or its purposes, but would still consent to the use of samples and data after they were introduced to the subject. Future research could demonstrate whether this is due to a high level of trust in the biobank or, for instance, a clear explanation of the biobank and its use.

Absolute refusal to the sharing of data with researchers outside Sanquin and other companies or organisations was very limited (respectively 7% and 2%). The finding that 62% of the donors would allow sharing both inside and outside of Europe, and 28% of donors only within the Netherlands corresponds with findings by the Netherlands Patients Federation [8]. However, their finding that 62% of the Dutch health care users would give consent to the use of the data for commercial purposes is much higher than the 7% that we found in our study [8]. This could be due to those participants being patients rather than healthy individuals, who may see benefits from health research with a commercial approach for

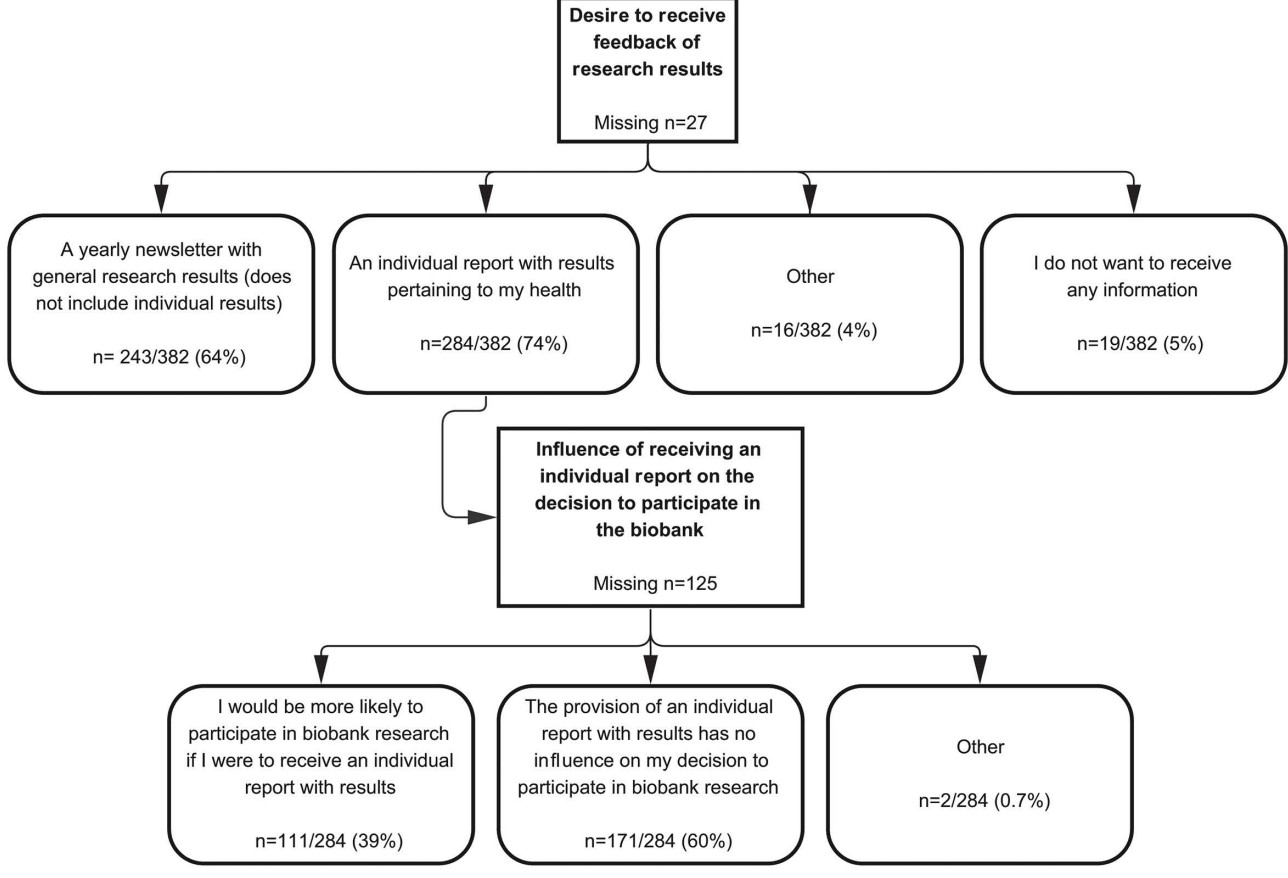

**Fig 2. The preference of participants in receiving feedback of research results, and whether or not receiving an individual report with results pertaining to their health would affect their decision to participate in the biobank.** Multiple answers were allowed for the question regarding feedback of research results.

themselves and other patients. In addition, donations are done on a voluntary non-remunerated and not-for-profit basis, which generally makes profitable activities in this setting ethically doubtful [15,16]. Commercial use of data is a recurring concern among participating donors to be hesitant to participate in the biobank. However, our research and previous research shows that commercial use can be acceptable if it has public benefit and there is sufficient trust in the organisation [17,18].

The absolute refusal by 25% of the donors to linking biobank data to data from other records, would limit the representativeness of studies for which linkage is required. Even though the percentage of participants that would give consent for the linkage to medical records (60%) is comparable to the 64.5% found by the previously mentioned study from the Gaza Strip [10], the donors' approval of linkage to information from registries, sources, and apps is lower (respectively 46%, 41%, and 10%). It is therefore recommendable that data linkage information is carefully explained, and data linking occurs carefully. As participants in this survey received less information on linkage compared to actual potential biobank participants, the approval rates could be an underestimation.

The importance of carefully explaining information is also shown by the willingness to storing DNA. Our study shows that 28% of the donors are willing to consent to storing of their DNA under certain conditions, as being informed about the usage of their DNA (66%) and receiving feedback on results about serious hereditary diseases (69%). Likewise, research

from Sweden [19] on reasons behind public refusal to DNA biobanking consent shows that extensive public information and deriving personal benefits would be prerequisites for future participation in DNA biobanking. Therefore, properly informing biobank participants concerning the terms and conditions of storing and using DNA in biobanking is necessary. Finally, the majority of donors (93%) would like to receive feedback on the research results, which corresponds with the findings of the Netherlands Patients Federation (73%), but with even more convincing numbers [8].

Strengths of this study include the quantitative data collection supplemented with open-ended questions and answer options, allowing for generalisation of the findings as well as in-depth insights into all themes. A limitation of this study may be a selection bias as people that respond to a survey received via a newsletter may also be more open to participate in other activities, such as the biobank itself. Also, there was overrepresentation of donors older than 56 years and from North Holland (including Amsterdam). The mean age of the active donor population in the Netherlands in 2009 was 48 for men and 42 for women [4], meaning that this overrepresentation may be a source of bias. On the other hand, the donor sample is large enough as determined a priori, and contains residents from all provinces of the Netherlands.

This study initially also aimed to include potential donors to research their attitude towards biobanking. Due to a low response of six non-donors, these responses were not included in this study. Future research could investigate whether non-donors show different attitudes than donors. Additionally, it would be interesting to look at potential differences between various types of donors, in terms of demographic as well as donation characteristics, and their willingness to give consent for a biobank.

In conclusion, this study has contributed to understanding donor attitudes towards storing blood samples and data for usage for research purposes, and all the related topics of data sharing, storage period, genetic research, and feedback options. The research findings show that donors are very willing to participate in a donor biobank. However, there also is a need for a proper introduction of the biobank subject and clear and comprehensive communication about the purposes and usage of the biobank. Concerns among donors could be reduced by well-informing them before, during and after the set-up and use of a biobank.

## Supporting information

**S1 File. Cross-sectional online survey to assess the attitude of donors to storing leftover blood materials and data in a biobank, data sharing, DNA storage, storage period and their opinions about potential feedback on research results.**
(DOCX)

**S2 File. Dataset with the results of the cross-sectional online survey.**
(XLSX)

## Acknowledgments

We would like to thank all the participants who filled out the survey.

## Author contributions

**Conceptualization:** Katja van den Hurk.

**Data curation:** Karlijn F. Joosten.

**Formal analysis:** Karlijn F. Joosten.

**Funding acquisition:** Katja van den Hurk.

**Investigation:** Lotte van Dammen.

**Methodology:** Lotte van Dammen.

**Supervision:** Lotte van Dammen, Katja van den Hurk.

**Validation:** Karlijn F. Joosten.

**Visualization:** Karlijn F. Joosten.

**Writing – original draft:** Karlijn F. Joosten.

**Writing – review & editing:** Karlijn F. Joosten, Lotte van Dammen, Eva-Maria Merz, Věra M. J. Novotný, Daphne C. Thijssen-Timmer, Katja van den Hurk.

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
