## [Decision Letter · Decision Letter 0]

6 Aug 2025

Dear Dr. Joosten,

Thank you for submitting your manuscript to PLOS ONE. After careful consideration, we feel that it has merit but does not fully meet PLOS ONE’s publication criteria as it currently stands. Therefore, we invite you to submit a revised version of the manuscript that addresses the points raised during the review process.

We look forward to receiving your revised manuscript.

Kind regards,

Sandar Tin Tin

Academic Editor

PLOS ONE

**Journal Requirements:**

1. When submitting your revision, we need you to address these additional requirements. Please ensure that your manuscript meets PLOS ONE's style requirements, including those for file naming. The PLOS ONE style templates can be found at https://journals.plos.org/plosone/s/file?id=wjVg/PLOSOne_formatting_sample_main_body.pdf and https://journals.plos.org/plosone/s/file?id=ba62/PLOSOne_formatting_sample_title_authors_affiliations.pdf 2. We note that this data set consists of interview transcripts. Can you please confirm that all participants gave consent for interview transcript to be published? If they DID provide consent for these transcripts to be published, please also confirm that the transcripts do not contain any potentially identifying information (or let us know if the participants consented to having their personal details published and made publicly available). We consider the following details to be identifying information:- Names, nicknames, and initials- Age more specific than round numbers- GPS coordinates, physical addresses, IP addresses, email addresses- Information in small sample sizes (e.g. 40 students from X class in X year at X university)- Specific dates (e.g. visit dates, interview dates)- ID numbers Or, if the participants DID NOT provide consent for these transcripts to be published:- Provide a de-identified version of the data or excerpts of interview responses- Provide information regarding how these transcripts can be accessed by researchers who meet the criteria for access to confidential data, including:a) the grounds for restrictionb) the name of the ethics committee, Institutional Review Board, or third-party organization that is imposing sharing restrictions on the datac) a non-author, institutional point of contact that is able to field data access queries, in the interest of maintaining long-term data accessibility.d) Any relevant data set names, URLs, DOIs, etc. that an independent researcher would need in order to request your minimal data set. For further information on sharing data that contains sensitive participant information, please see: https://journals.plos.org/plosone/s/data-availability#loc-human-research-participant-data-and-other-sensitive-data If there are ethical, legal, or third-party restrictions upon your dataset, you must provide all of the following details (https://journals.plos.org/plosone/s/data-availability#loc-acceptable-data-access-restrictions):a) A complete description of the datasetb) The nature of the restrictions upon the data (ethical, legal, or owned by a third party) and the reasoning behind themc) The full name of the body imposing the restrictions upon your dataset (ethics committee, institution, data access committee, etc)d) If the data are owned by a third party, confirmation of whether the authors received any special privileges in accessing the data that other researchers would not havee) Direct, non-author contact information (preferably email) for the body imposing the restrictions upon the data, to which data access requests can be sent 3. We note that there is identifying data in the Supporting Information file. Due to the inclusion of these potentially identifying data, we have removed this file from your file inventory. Prior to sharing human research participant data, authors should consult with an ethics committee to ensure data are shared in accordance with participant consent and all applicable local laws. Data sharing should never compromise participant privacy. It is therefore not appropriate to publicly share personally identifiable data on human research participants. The following are examples of data that should not be shared: -Name, initials, physical address-Ages more specific than whole numbers-Internet protocol (IP) address-Specific dates (birth dates, death dates, examination dates, etc.)-Contact information such as phone number or email address-Location data-ID numbers that seem specific (long numbers, include initials, titled “Hospital ID”) rather than random (small numbers in numerical order) Data that are not directly identifying may also be inappropriate to share, as in combination they can become identifying. For example, data collected from a small group of participants, vulnerable populations, or private groups should not be shared if they involve indirect identifiers (such as sex, ethnicity, location, etc.) that may risk the identification of study participants. Additional guidance on preparing raw data for publication can be found in our Data Policy (https://journals.plos.org/plosone/s/data-availability#loc-human-research-participant-data-and-other-sensitive-data) and in the following article: http://www.bmj.com/content/340/bmj.c181.long. Please remove or anonymize all personal information (<specific identifying information in file to be removed>), ensure that the data shared are in accordance with participant consent, and re-upload a fully anonymized data set. Please note that spreadsheet columns with personal information must be removed and not hidden as all hidden columns will appear in the published file. 4. Please upload a new copy of Figures 1 and 2, as the detail is not clear. Please follow the link for more information: "https://blogs.plos.org/plos/2019/06/looking-good-tips-for-creating-your-plos-figures-graphics/"  5. https://blogs.plos.org/plos/2019/06/looking-good-tips-for-creating-your-plos-figures-graphics/" 5. If the reviewer comments include a recommendation to cite specific previously published works, please review and evaluate these publications to determine whether they are relevant and should be cited. There is no requirement to cite these works unless the editor has indicated otherwise. 

Reviewers' comments:

Reviewer's Responses to Questions

**Comments to the Author**

1. Is the manuscript technically sound, and do the data support the conclusions?

Reviewer #1: Yes

Reviewer #2: Yes

2. Has the statistical analysis been performed appropriately and rigorously?

Reviewer #1: Yes

Reviewer #2: No

3. Have the authors made all data underlying the findings in their manuscript fully available?

Reviewer #1: Yes

Reviewer #2: Yes

4. Is the manuscript presented in an intelligible fashion and written in standard English?

Reviewer #1: Yes

Reviewer #2: Yes

**Reviewer #1:**  The manuscript by Joosten and co-workers describes attitudes of blood donors towards biobanking in the Netherlands. Blood donor population has been suggested, and proved as well, to be an excellent source of biosamples from (relatively) healthy individuals. They find that a vast majority of donors are willing to consent to biobanking. The study is interesting and important addition to the current knowledge. There are a few issues I hope the authors would consider:

The survey was announced in the donor new letter and carried out July-August. Table 1 shows the basic demographics of the 409 participants. How well this population represents the overall donor population? They could show the figures of the overall donor population for comparison (perhaps no statistical comparison needed as the populations sizes are so different). I would assume that the participants were older, frequent donors who are willing to help in many different ways. There is a short discussion on the topic but the authors might raise this issue more as a possible bias.

It is interesting, or a bit surprising, that the majority of the donors did not know what biobanking means but still would be willing to give samples and data. The authors could discuss is it so that the trust to blood bank (here the Sanquin) is so strong that basically the (frequent, committed) donors would be willing to almost anything asked?

In the discussion, lines 250-251 the authors state that the sample of 400 was estimated a priori to be large enough for meaningfull analyses. They should describe in e.g. Method how this was estimated.

Minor points:

They could cite Mitchell, R. Blood banks biobanks and the ethics of donation. Transfusion (Paris) 50, 1866–1899 (2010), an editorial in which use of donor samples in biobanks was discussed.

Tthey might add some citations of large-scale biobank studies, such as the FinnGen (Kurki, M. I. et al. FinnGen provides genetic insights from a well-phenotyped isolated population. Nature 613, 508–519, 2023), in which biobanked blood donor samples have been used.

An asset of samples from healthy individuals in disease-genetics studies is that, as being from healthy individuals, the disease progession or treatments do not confound the effects of genetic variants of possible interest. They hence may be an interesting option for molecular studies. The authors might think about this in the Discussion.

**Reviewer #2: ** More information is needed about participant selection. The authors stated in the methodology that the survey invitation was sent to a randomly selected group. How was this done? Did the study use a sample selection software?

Data analysis limited to percentages is insufficient. A more in-depth analysis involving cross-tabulations or logistic regression is required.

**Do you want your identity to be public for this peer review?** For information about this choice, including consent withdrawal, please see our Privacy Policy

Reviewer #1: **Yes: ** Jukka Partanen

Reviewer #2: No

---

## [Author Response · Author response to Decision Letter 1]

11 Sep 2025

Dear Dr. Sandar Tin Tin,

Thank you for your consideration of the manuscript ‘Attitudes of blood and plasma donors towards biobanking for longitudinal health research’ and supplying us with the needed improvements for the journal requirements as well as the reviewers’ comments.

Journal Requirements:

Please ensure that your manuscript meets PLOS ONE's style requirements, including those for file naming. The PLOS ONE style templates can be found at https://journals.plos.org/plosone/s/file?id=wjVg/PLOSOne_formatting_sample_main_body.pdf and https://journals.plos.org/plosone/s/file?id=ba62/PLOSOne_formatting_sample_title_authors_affiliations.pdf

We would like to thank the editor for this overview of PLOS ONE style requirements, with the style templates as clear and helpful examples.

We have adjusted the title page (page 1) by removing the second title (line 7, page 1), adding the asterix to the corresponding author (line 14, page 1), reordering the components in the affiliations from small to large (line 16, line 19, line 21, page 1), and replacing the address of the corresponding author by the email address and initials in parentheses (line 26 to line 29, page 1).

We also adjusted the font of the level 1 and level 2 headings throughout the manuscript, in accordance with the template of the main body (line 33 and line 52, page 2; line 94, line 95 and line 100, page 4; line 112 and line 127, page 5; line 139, line 148 and line 149, page 6; line 155, page 7; line 174, page 8; line 193, page 9; line 201 and line 212, page 10; line 275, page 12; line 277, page 13; line 329, page 15), the italics in the headings were removed (line 95 and line 100, page 4; line 112 and line 127, page 5; line 139 and line 149, page 6; line 155, page 7; line 174, page 8; line 193, page 9; and line 201, page 10), and headings were written in sentence case (line 94 and line 95, page 4; line 112, page 5; line 139, page 6).

The figure captions were made bold and not cursive (line 198-200 and line 209-211, page 10) and the table captions were made bold instead of cursive (line 154, page 6; line 163, page 7; line 192, page 9). Likewise, the supporting information titles were renamed to ‘S1 File’ and ‘S2 File’ and were made bold (line 119, page 5; line 330-332 and line 333, page 15). The empty rows to create spacing in the tables were removed (line 163, page 7; line 192, page 9). Finally, an underscore was added in the file naming of the two Supporting information files.

2. We note that this data set consists of interview transcripts. Can you please confirm that all participants gave consent for interview transcript to be published?

If they DID provide consent for these transcripts to be published, please also confirm that the transcripts do not contain any potentially identifying information (or let us know if the participants consented to having their personal details published and made publicly available). We consider the following details to be identifying information:

- Names, nicknames, and initials

- Age more specific than round numbers

- GPS coordinates, physical addresses, IP addresses, email addresses

- Information in small sample sizes (e.g. 40 students from X class in X year at X university)

- Specific dates (e.g. visit dates, interview dates)

- ID numbers

Or, if the participants DID NOT provide consent for these transcripts to be published:

- Provide a de-identified version of the data or excerpts of interview responses

- Provide information regarding how these transcripts can be accessed by researchers who meet the criteria for access to confidential data, including:

a) the grounds for restriction

b) the name of the ethics committee, Institutional Review Board, or third-party organization that is imposing sharing restrictions on the data

c) a non-author, institutional point of contact that is able to field data access queries, in the interest of maintaining long-term data accessibility.

d) Any relevant data set names, URLs, DOIs, etc. that an independent researcher would need in order to request your minimal data set.

For further information on sharing data that contains sensitive participant information, please see: https://journals.plos.org/plosone/s/data-availability#loc-human-research-participant-data-and-other-sensitive-data

If there are ethical, legal, or third-party restrictions upon your dataset, you must provide all of the following details (https://journals.plos.org/plosone/s/data-availability#loc-acceptable-data-access-restrictions):

a) A complete description of the dataset

b) The nature of the restrictions upon the data (ethical, legal, or owned by a third party) and the reasoning behind them

c) The full name of the body imposing the restrictions upon your dataset (ethics committee, institution, data access committee, etc)

d) If the data are owned by a third party, confirmation of whether the authors received any special privileges in accessing the data that other researchers would not have

e) Direct, non-author contact information (preferably email) for the body imposing the restrictions upon the data, to which data access requests can be sent

We express our gratitude to the editor for this point of advice, as well as the extensiveness of the included information on this topic. We think there has been a misunderstanding, as the data set does not contain interview transcripts: these are respondents’ typed responses in the free-text fields linked to the questions in the survey. Furthermore, the open answers were coded to prevent any information to be potentially identifying. With this information, could you please indicate whether anything still needs to be added regarding the consent description and how we have included the (coded) answers to the free-text fields in the dataset?

3. We note that there is identifying data in the Supporting Information file. Due to the inclusion of these potentially identifying data, we have removed this file from your file inventory. Prior to sharing human research participant data, authors should consult with an ethics committee to ensure data are shared in accordance with participant consent and all applicable local laws.

-Location data

We would like to thank the editor for her sharpness on this point of privacy, as that is very important when working with participant data. We have thoroughly investigated this point of the editor but we could not find any of the named examples of data that should not be shared in the Supporting Information file, neither any combinations of data that can become identifying. In our view the groups are also large enough to make them unidentifiable. Moreover, provinces in the Netherlands are big areas, meaning that the provinces data would not lead to identification. Nevertheless, if the locations where the respondents usually donate are the problem in this case, we could remove that specific data from the Supporting Information file. Could you please clarify which data in the Supporting Information file is considered identifying data? We would like to thank you in advance for your time.

4. Please upload a new copy of Figures 1 and 2, as the detail is not clear. Please follow the link for more information: "https://blogs.plos.org/plos/2019/06/looking-good-tips-for-creating-your-plos-figures-graphics/" https://blogs.plos.org/plos/2019/06/looking-good-tips-for-creating-your-plos-figures-graphics/"

We would like to thank the editor for the provided link about clear figures. However, we do not understand the problem because they are both vector files which should have no resolution or detail issues. We could not see the problem of no clear details in our figures. We provided extra .tif versions of the figures, using the suggested Preflight Analysis and Conversion Engine (PACE) digital diagnostic tool, that we were hoping that would solve the problem. However, these .tif versions generated by the PACE tool had major problems in resolution. Therefore, we uploaded our .eps files once again, hoping that these are sufficient to meet the requirements of the figure resolution.

We would like to thank the editor for this valuable addition to the reviewers’ recommendations, which allowed us to openly look at the recommendations without feeling obliged to include them.

We would like to thank the editor for this recommendation. We carefully and thoroughly checked the reference list and made some small adjustments in the references itself to make sure that the exact reference was used (line 314, line 323 and line 325, page 14). Furthermore, 2 citations were added based on the reviewers’ suggestions (line 281-283, page 13; line 319-320, page 14). In accordance, the reference numbers in the square brackets were adjusted through the whole text.

Additionally, the mentioning of a study in the text was adjusted from research/study ‘by author name et al. (year)’ to ‘from country’ (line 242 and line 250, p11).

Reviewers' comments:

Reviewer's Responses to Questions

Comments to the Author

1. Is the manuscript technically sound, and do the data support the conclusions?

Reviewer #1: Yes

Reviewer #2: Yes

2. Has the statistical analysis been performed appropriately and rigorously?

Reviewer #1: Yes

Reviewer #2: No

3. Have the authors made all data underlying the findings in their manuscript fully available?

Reviewer #1: Yes

Reviewer #2: Yes

4. Is the manuscript presented in an intelligible fashion and written in standard English?

Reviewer #1: Yes

Reviewer #2: Yes

5. Review Comments to the Author

Reviewer #1: The manuscript by Joosten and co-workers describes attitudes of blood donors towards biobanking in the Netherlands. Blood donor population has been suggested, and proved as well, to be an excellent source of biosamples from (relatively) healthy individuals. They find that a vast majority of donors are willing to consent to biobanking. The study is interesting and important addition to the current knowledge. There are a few issues I hope the authors would consider:

The survey was announced in the donor new letter and carried out July-August. Table 1 shows the basic demographics of the 409 participants. How well this population represents the overall donor population? They could show the figures of the overall donor population for comparison (perhaps no statistical comparison needed as the populations sizes are so different). I would assume that the participants were older, frequent donors who are willing to help in many different ways. There is a short discussion on the topic but the authors might raise this issue more as a possible bias.

We would sincerely like to thank Jukka Partanen for the review and the provided suggestions.

The suggestion to dive deeper in the comparison of this population with the overall donor population has been seriously considered. We think that Jukka Partanen is right in her assumption that people who respond to the survey are different from the general donor population. But at the same time, it is difficult to prove this, because there is little published data available on the precise characteristics of the general donor population in the Netherlands. At this point, the only comparison we can make is between the mean age of men and women of the active donor population in the Netherlands in 2009, from the research by Timmer et al. (2019), and the age categories we have in our survey [4]. This suggests that the mean age of the survey population is slightly older than the mean age of the donor population. We added this to the discussion, with a note that this may indeed be a source of bias (line 260-262, page 12).

It is interesting, or a bit surprising, that the majority of the donors did not know what biobanking means but still would be willing to give samples and data. The authors could discuss is it so that the trust to blood bank (here the Sanquin) is so strong that basically the (frequent, committed) donors would be willing to almost anything asked?

The suggestion to discuss the fact that the majority of donors did not know w

---

## [Decision Letter · Decision Letter 1]

25 Nov 2025

Attitudes of blood and plasma donors towards biobanking for longitudinal health research

PONE-D-25-19043R1

Dear Dr. Joosten,

We’re pleased to inform you that your manuscript has been judged scientifically suitable for publication and will be formally accepted for publication once it meets all outstanding technical requirements.

Kind regards,

Xiang Zhu

Academic Editor

PLOS ONE

Additional Editor Comments (optional):

Reviewers' comments:

Reviewer's Responses to Questions

**Comments to the Author**

Reviewer #1: (No Response)

Reviewer #2: All comments have been addressed

2. Is the manuscript technically sound, and do the data support the conclusions?

Reviewer #1: Yes

Reviewer #2: Yes

3. Has the statistical analysis been performed appropriately and rigorously?

Reviewer #1: Yes

Reviewer #2: Yes

4. Have the authors made all data underlying the findings in their manuscript fully available?

Reviewer #1: Yes

Reviewer #2: Yes

5. Is the manuscript presented in an intelligible fashion and written in standard English?

Reviewer #1: Yes

Reviewer #2: No

Reviewer #1: I wantn to thank the authors for adequate response to my comments and for the revised version of the manuscript.

Reviewer #2: The English needs to be improved. For example:

Line 82, page 3: change “12 300 questionnaire…” to “12,300 questionnaire ..”. This should be done to similar numbers across manuscript.

Line 85, add “3” after “Within the Netherlands”

**Do you want your identity to be public for this peer review?** For information about this choice, including consent withdrawal, please see our Privacy Policy

Reviewer #1: **Yes: ** Jukka Partanen

Reviewer #2: No

---

## [Editor Report · Acceptance letter]

PONE-D-25-19043R1

PLOS One

Dear Dr. Joosten,

I'm pleased to inform you that your manuscript has been deemed suitable for publication in PLOS One. Congratulations! Your manuscript is now being handed over to our production team.

Kind regards,

on behalf of

Dr. Xiang Zhu

Academic Editor

PLOS One